# In Vitro Functional Validation of an Anti-FREM2 Nanobody for Glioblastoma Cell Targeting

**DOI:** 10.3390/antib14010008

**Published:** 2025-01-24

**Authors:** Gloria Krapež, Neja Šamec, Alja Zottel, Mojca Katrašnik, Ana Kump, Jernej Šribar, Igor Križaj, Jurij Stojan, Rok Romih, Gregor Bajc, Matej Butala, Serge Muyldermans, Ivana Jovčevska

**Affiliations:** 1Center for Functional Genomics and Biochips, Institute of Biochemistry and Molecular Genetics, Faculty of Medicine, University of Ljubljana, Zaloška cesta 4, 1000 Ljubljana, Slovenia; gloria.krapez@mf.uni-lj.si (G.K.); neja.samec@mf.uni-lj.si (N.Š.); alja.zottel@mf.uni-lj.si (A.Z.); mojca.katrasnik@mf.uni-lj.si (M.K.); 2Department of Molecular and Biomedical Sciences, Jožef Stefan Institute, Jamova 39, 1000 Ljubljana, Slovenia; ana.kump16@gmail.com (A.K.); igor.krizaj@ijs.si (I.K.); 3Jožef Stefan International Postgraduate School, Jamova 39, 1000 Ljubljana, Slovenia; 4Institute of Biochemistry and Molecular Genetics, Faculty of Medicine, University of Ljubljana, Vrazov trg 2, 1000 Ljubljana, Slovenia; jurij.stojan@mf.uni-lj.si; 5Institute of Cell Biology, Faculty of Medicine, University of Ljubljana, Vrazov trg 2, 1000 Ljubljana, Slovenia; rok.romih@mf.uni-lj.si; 6Department of Biology, Biotechnical Faculty, University of Ljubljana, Jamnikarjeva 101, 1000 Ljubljana, Slovenia; gregor.bajc@bf.uni-lj.si (G.B.); matej.butala@bf.uni-lj.si (M.B.); 7Cellular and Molecular Immunology, Vrije Universiteit Brussel, Pleinlaan 2, 1050 Brussels, Belgium

**Keywords:** brain cancer, membrane-bound protein, cytotoxicity, molecular tool, subcellular localization

## Abstract

**Background/Objectives**: Glioblastomas are the most common brain malignancies. Despite the implementation of multimodal therapy, patient life expectancy after diagnosis is barely 12 to 18 months. Glioblastomas are highly heterogeneous at the genetic and epigenetic level and comprise multiple different cell subpopulations. Therefore, small molecules such as nanobodies, able to target membrane proteins specific to glioblastoma cells or specific cell types within the tumor are being investigated as novel tools to treat glioblastomas. **Methods**: Here, we describe the identification of such a nanobody and its in silico and in vitro validation. NB3F18, as we named it, is directed against the membrane-associated protein FREM2, overexpressed in glioblastoma stem cells. **Results**: Three dimensional in silico modeling indicated that NB3F18 and FREM2 form a stable complex. Surface plasmon resonance confirmed their interaction with moderate affinity. As we demonstrated by flow cytometry, NB3F18 binds to glioblastoma stem cells to a greater extent than to differentiated glioblastoma cells and astrocytes. Immunocytochemistry revealed surface localization of NB3F18 on glioblastoma stem cells, whereas cytoplasmic localization of NB3F18 was observed in other cell lines. NB3F18 was detected by transmission electron microscopy on the plasma membrane and in various compartments of the endocytic pathway, from endocytic vesicles to multivesicular bodies (endosomes) and lysosomes. Interestingly, NB3F18 was cytotoxic to glioblastoma stem cells. **Conclusions**: Collectively, NB3F18 has been qualified as an interesting tool to target glioblastoma cells and as a potential vehicle to deliver biological or pharmaceutical agents to these cells.

## 1. Introduction

Gliomas account for nearly 80% of primary brain malignancies [1], 60–70% of which are glioblastomas [2,3]. At the time of diagnosis, patients with primary glioblastoma are on average 64 years old [4,5], whereas secondary glioblastomas are diagnosed in adults aged 45 years or younger [6]. Even with our comprehensive understanding of the genetic diversity of glioblastoma as originally reported [7,8] and complemented by Verhaak et al. [9], prevention and treatment of glioblastoma is one of the major challenges of neuro-oncology.

Despite aggressive clinical care that follows the Stupp protocol [10], patients usually succumb to the disease within 12 to 18 months [11]. Glioblastoma is uniformly treated as a single disease even though molecular profiling has shown that glioblastomas are heterogeneous and comprise many cell types with different characteristics [12]. The diversity of clonal and subclonal differentiated tumor cell populations, glioblastoma stem cells (GSCs), and multiple non-tumor cells, such as endothelial and inflammatory cells, and other components of the tumor microenvironment all contribute to the heterogeneity. This results in varied genetic and protein profiles that participate to differences in treatment response and patient outcomes [13,14,15]. Furthermore, GSCs play an indispensable role in the formation, maintenance, and recurrence of heterogeneous glioblastomas that resemble the original parent tumor, indicating that GSCs are a crucial target for treatment. The use of various specific surface markers or molecular mediators such as prominin-1 (CD133), cluster of differentiation 90 (CD90), cell surface glycoprotein CD44 (CD44), L1 cell adhesion molecule (L1CAM), and glycerol-3-phosphate dehydrogenase 1 (GPD1) is a common approach to identify GSCs and define lineage-specific subpopulations within the tumor [16] and also offers new directions for the development of modern therapies. Moreover, currently available treatment only partially targets the heterogeneous populations of glioblastoma cells, whereas the resistant cancer cells and GSCs are left behind untreated, given ample time to recover from the initial treatment [17,18]. In 80–90% of the cases, this leads to tumor recurrence with a more aggressive phenotype [19]. Molecules targeting specific cell subsets, for example aimed towards GSC markers including CD133 [20,21], epidermal growth factor receptor (EGFR) [22,23,24], sonic hedgehog protein (Shh) [25], and signal transducer and activator of transcription 3 (STAT3) [26], as well as related signaling pathways, are being investigated as such an approach is expected to yield better results. For example, CAR-T therapies targeting CD133, which is closely associated with tumorigenicity, therapy resistance, and self-renewal, have shown promise in preclinical models [27]. Similarly, novel EGFR inhibitors and EGFRvIII-targeted therapies such as vaccines are actively being studied as EGFR, which is frequently overexpressed or mutated in glioblastoma, drives tumor proliferation and resistance [28]. Targeting the Shh pathway, known to support GSC maintenance and proliferation, with inhibitors like vismodegib was examined in a clinical trial [29]. Similarly, the oral p-STAT3 inhibitor WP1066 was tested in a Phase I clinical trial [30]. For successful implementation in clinical care, these cell-targeting molecules should be biocompatible, biodegradable, non-toxic, stable after administration, and easily produced on a large scale with controllable physical and chemical properties [31].

The antigen-binding fragments of heavy-chain antibodies naturally occurring in the serum of camelids [32] fulfil these criteria. These antigen-binding fragments, termed nanobodies, consist of a prolate-shaped, single domain of small molecular mass (14 kDa) and dimensions in the single digit nanometer range [33]. Just like variable domains of classical immunoglobulins, nanobodies comprise four framework regions (FRs) and three complementarity-determining regions (CDRs). With amino acid sequence identities of approximately 80%, nanobodies are considered structurally similar to classical immunoglobulins. This makes them low or even non-immunogenic and suitable for use in patients [34]. Nanobodies have been used as molecular tools for modulating protein function and studying the role of proteins of interest in cells [35,36,37].

Starting as research tools, nanobodies were first translated in clinical settings in 2018 and 2019 with the approval of the bivalent nanobody Caplacizumab for treatment of patients with thrombotic thrombocytopenic purpura (TTP) [38] by the European Medicines Agency (EMA) and the U.S. Food and Drug Administration (FDA), respectively. Since then, three more therapeutic nanobodies received approval: (i) in 2020 and 2021, the FDA and China’s National Medical Products Administration (NMPA) approved the anti-PD-L1 humanized nanobody Envafolimab for the treatment of various solid tumors [39,40,41]; (ii) in 2022, the FDA and EMA approved the nanobody-based CAR T-cell therapy Carvytki for the treatment of relapsed or refractory multiple myeloma [42]; and (iii) in 2022, the Pharmaceuticals and Medical Devices Agency (PMDA) of Japan approved Ozoralizumab, a bivalent trispecific nanobody, for the treatment of rheumatoid arthritis [43]. With approvals spanning treatments for TTP, solid tumors, multiple myeloma, and rheumatoid arthritis, nanobodies show potential for wide use in clinical care.

In our previous research [44,45,46] we identified and validated the *FRAS1-related extracellular matrix 2* (*FREM2*) gene and protein as a specific component of glioblastomas in silico, in vitro, and in human tissue samples. *FREM2* encodes for a transmembrane protein localized to the cell basement membrane that is in direct contact with integrins, eukaryotic translation initiation factor 6 (EIF6), and many other membrane receptors involved in cellular migration and motility [47,48]. The FREM2 transmembrane protein is part of the FRAS1/FREM complex. Its main role is mediation between the FREM1 and FRAS1 proteins and with that the formation of a stable basement membrane. In our previously published research [45], *FREM2* was found to be specific for glioblastoma cells, i.e., overexpressed in GSCs compared to differentiated glioblastoma cells at the gene and protein levels. Extended analysis using TCGA data confirmed in silico and experimentally the differential expression of FREM2 at the gene and protein levels [44]. Recently, it has also been shown that observing changes in the FREM2 signaling pathway, rather than only the *FREM2* expression level, represents a more reliable prognostic glioblastoma biomarker that is comparable to the already established isocitrate dehydrogenase (*IDH*) biomarker [46]. Moreover, it is correlated with low progression-free survival in glioblastomas.

The objective of this research was to obtain and functionally validate a nanobody able to selectively target glioblastoma cells. We selected the nanobody NB3F18 targeting FREM2, modeled its complex, and determined the binding affinity. We demonstrate the ability of NB3F18 to bind to the surface of glioblastoma stem cells in vitro and present its subcellular localization. At last, we show that NB3F18 can inhibit proliferation of NCH644 glioblastoma stem cells. The nanobody characterized in this study can be used as a robust molecular tool to analyze the role of FREM2 in glioblastoma initiation and progression in vitro and potentially in vivo.

## 2. Materials and Methods

### 2.1. Cell Cultures

Four glioblastoma cell lines and one reference cell line were used. NCH644 (300124) and NCH421K (300118) glioblastoma stem-like cell lines were obtained from the Cell Line Service, Eppelheim, Germany. U251MG (09063001, Sigma Aldrich, St. Louis, MO, USA) and U87MG (HTB-14) cell lines were obtained from the American Type Culture Collection (ATCC), Manassas, VA, USA. Human astrocytes (1800) were obtained from ScienCell, Carlsbad, CA, USA.

All cell lines were maintained in a humidified cell incubator at 37 °C and 5% CO_2_. Glioblastoma stem-like cell lines NCH644 and NCH421K were expanded in neurobasal medium (21103-049, Gibco, Waltham, MA, USA) supplemented with 1% GlutaMAX (35050-038, Gibco), 1% antibiotic/antimycotic (15240-062, Gibco), 2% B-27 supplement (17504-044, Gibco), 20 ng/mL EGF (PHG0314, Gibco), 20 ng/mL bFGF (PHG0024, Gibco), and 1 U/mL heparin (H3149, Sigma Aldrich). The glioblastoma cell lines U251MG and U87MG were expanded in minimum essential medium Eagle (M4655, Sigma Aldrich) supplemented with 10% fetal bovine serum (F9665, Sigma Aldrich), antibiotic/antimycotic (15240-062, Gibco), sodium pyruvate (S8636, Sigma Aldrich), and nonessential amino acids (11140-035, Gibco). For seeding human astrocytes, cell culture flasks were pre-treated with poly-D-lysine (P6407, Sigma Aldrich). Human astrocytes were expanded in astrocyte medium (1801, ScienCell) supplemented with astrocyte growth supplement (1852, ScienCell), penicillin/streptomycin (0503, ScienCell), and fetal bovine serum (0010, ScienCell). All of the cell lines were regularly tested for Mycoplasma using Venor^®^GeM Classic (Minerva Biolabs, Berlin, Germany).

### 2.2. FREM2 Recombinant Protein Fragment

We used the recombinant protein fragment of human FREM2 (APrEST72066, Atlas antibodies, Bromma, Sweden). To ensure protein immobilization through the His-tag present in the recombinant protein, we used Pierce™ Nickel Coated Plates (15442, Pierce, Waltham, MA, USA).

### 2.3. Library Enrichment and Screening

An available nanobody immune library [49] was used for the experiments. The nanobody library was obtained by repeatedly immunizing an adult alpaca with whole glioma cells enriched in GSCs. The glioma cells were isolated from a patient tumor tissue sample and grown on laminin under conditions allowing for the enrichment of glioblastoma stem cells as described by Bourkoula et al. [50]. The animal was immunized 7 times at one-week intervals. Each immunization consisted of subcutaneously injecting the animal with 10^6^ whole cells resuspended in one milliliter phosphate-buffered saline (PBS). After the last immunization, 50 mL blood was drawn from the animal and used for lymphocyte purification and library construction in pHEN4 [51]. The library consisted of approximately 10^8^ individual transformants. Phage enrichment was performed as described by Hassanzadeh-Ghassabeh et al. [51]. Briefly, to collect phages with high affinity to their target molecule [52], the recombinant human FREM2 was immobilized on nickel-coated microtiter plates at a concentration of 100 µg/mL. Library phages were incubated with immobilized protein and extensively washed to remove phages devoid of target recognizing nanobodies. Phage particles expressing target-specific nanobodies were eluted with trimethylamine solution and used to infect fresh *E. coli* cells. Bacterial cultures were grown overnight; phage particles from the culture supernatant were precipitated with PEG-6000/2.5 M NaCl and used in the next round of panning. After the third round of panning, we proceeded to the screening of individual colonies.

Single colonies, obtained after each round of panning, were used for screening the proteins of their periplasmic extract in an enzyme-linked immunosorbent assay (ELISA). Individual clones were inoculated in 1 mL terrific broth (TB) medium and incubated for 4 to 5 h. Nanobody expression was induced by isopropyl β-D-1-thiogalactopyranoside (IPTG, I6758, Sigma Aldrich) at 1 mM final concentration. On the next day, osmotic shock was generated by resuspending the packed cells in an equal volume of a hypertonic (Tris/EDTA/sucrose) solution and incubation for 1 h at 4 °C, followed by the addition of two volumes distilled water and 2 h incubation at 4 °C. Periplasmic proteins present in the supernatant were applied to wells coated with FREM2 and to control wells (without FREM2). Nanobodies were detected through the presence of their hemagglutinin (HA)-tag with primary (H3663, Sigma Aldrich) and secondary (A3562, Sigma Aldrich) antibodies. After application of the alkaline phosphatase substrate (P4744, Sigma Aldrich), optical density was measured on a Synergy H4 Hybrid (BioTek, Shoreline, VT, USA) at 405 nm at different time intervals.

### 2.4. Nanobody Selection

Nanobody genes of the ELISA-positive clones were amplified by colony PCR (6 min at 95 °C, 45 s at 94 °C, 45 s at 55 °C, 45 s at 72 °C, and 10 min at 72 °C) using RP (5′**→**3′ TCA CAC AGG AAA CAG CTA TGA C) and GIII (5′**→**3′ CCA CAG ACA GCC CTC ATA G) primers. Fragments of appropriate size were sequenced at Macrogen (Amsterdam, The Netherlands). Nanobody sequences were analyzed with MEGA 5.2 software [53]. Parameters of the selected nanobody were computed with the Expasy ProtParam tool [54].

### 2.5. Subcloning, Large-Scale Expression, and Purification

Procedures were performed as previously described [55,56]. First, the nanobody was subcloned from the pHEN4 to the pHEN6 vector. The replacement of the genes for the *HA-tag* by that of the *His_6_-tag* and the deletion of the *gene3p* are the main differences between pHEN4 and pHEN6. Single colonies were amplified with colony PCR using the same settings as described in the Nanobody Selection section, using A6E (5′**→**3′ GAT GTG CAG CTG CAG GAG TCT GGR GGA GG) and 38 (5′**→**3′ GGA CTA GTG CGG CCG CTG GAG ACG GTG ACC TGG GT) primers. PCR amplicons were purified with the GenElute™ PCR Clean-Up Kit (NA1020, Sigma Aldrich) and digested overnight at 37 °C with *Pst*I (10 U/μL, ER0611) and *Eco91*I (*BstE*II) (10 U/μL, ER0392), both from Thermo Fisher Scientific, Stockholm, Sweden. The next day, digested products were ligated to the previously digested pHEN6 vector. Ligation was performed for 2 h at room temperature. Ligation mixtures were transformed into *E. coli* WK6 cells and plated on Luria–Bertani (LB) agar plates with ampicillin and glucose. Nanobody inserts from single colonies were amplified by colony PCR (RP and FP (5′**→**3′ CGC CAG GGT TTT CCC AGT CAC GAC) primers) and sequenced.

Single colonies were grown overnight in LB medium supplemented with ampicillin. They were used to inoculate TB medium, and cells were cultured for 4 to 5 h. Nanobody expression was induced by adding IPTG to a final concentration of 10 µM and overnight incubation. The next day, periplasmic extracts were obtained by osmotic shock and incubated overnight at 4 °C with Ni^2+^-nitrilotriacetic acid agarose (30210, Qiagen, Hilden, Germany). After washing non-bound material, captured nanobodies were eluted with a one-column volume of 0.5 M imidazole in PBS. Nanobodies were purified with immobilized metal affinity chromatography (IMAC) and size exclusion chromatography (SEC) on an ÄKTA purifier (GE Healthcare, Chicago IL, USA). Nanobody elution was followed by measuring the optical density at 280 nm, from which the nanobody concentration was calculated taking into account the extinction coefficient, using the Unicorn 7.3 software (Cytvia, Marlborough, MA, USA). The purity of the obtained nanobody was assessed on a 4–12% Bis-Tris mini protein gel (NP0321BOX, Invitrogen, Waltham, MA, USA). After electrophoresis, the gel was stained with Coomassie brilliant blue and destained overnight in destaining solution (10% methanol, 40% acetic acid, and 50% distilled water).

### 2.6. Affinity Measurements and Analysis of Binding

Surface plasmon resonance (SPR) assays were performed on a Biacore T200 (GE Healthcare) apparatus at 25 °C. FREM2 or vimentin (as a negative control) proteins were immobilized via free amino groups using the carboxymethylated dextran CM5 sensor chip (29-1049-88, Cytvia). Before covalent protein immobilization, the chip surface was activated with 0.4 M 1-ethyl-3-(3-dimethylaminopropyl)-carbodiimide hydrochloride and 0.1 M N-hydroxysuccinimide. Injection of FREM2 or vimentin was adjusted to obtain 990 (flow-cell 2) or 1080 (flow-cell 4) response units of immobilized protein, respectively. The running buffer for the SPR assays was 2.7 mM KCl, 8 mM Na_2_HPO_4_, 2 mM KH_2_PO_4_ (pH 7.4), 137 mM NaCl, 3 mM EDTA, and 0.005% P20. NB3F18 was diluted in running buffer and injected over chip-immobilized proteins in concentrations ranging from 9.76 to 2500 nM. NB3F18 was injected at a flow rate of 30 µL per minute for 60 s, and the dissociation was followed for 60 s. The chip surface was regenerated with 2 mM NaOH for 30 s followed by a 90 s stabilization period and a 30 s injection of running buffer. Sensorgrams were corrected for corresponding responses of flow-cell 1 without immobilized proteins. Biacore T200 evaluation software Version: 3.2.1 was used to determine the equilibrium dissociation constant by fitting data to a steady-state affinity model. Experiments were performed in triplicates.

### 2.7. Visualization of 3D Interaction Between Nanobody NB3F18 from Vicuña Pacos and Human FREM2

We developed a 3D model to visualize the interaction complex of nanobody NB3F18 and the binding epitope of FREM2. Since the 3D structure of neither the nanobody nor FREM2 has been solved, we searched for suitable homology building templates. We found very high protein sequence identity (63%) between NB3F18 and a camelid nanobody from a solved protein–protein complex in the Protein Data Bank (PDB) entry 5o0w [57]. The sequence of NB3F18 was submitted to the Swiss-model WEB suite [58] for homology building, using chain E from PDB entry 5o0w as a template. The resulting 3D model was subjected to a short optimization/relaxation run with the Chemistry at Harvard Macromolecular Mechanics (CHARMM) molecular simulation program [59].

The portion of FREM2 protein that interacts with NB3F18 also had to be modeled by homology. An existing model structure of FREM2 (UniProtKB code Q5SZK8 or its secondary accession number Q4QQG1) was found in the Modeller’s database, developed by Sali’s lab [60] under the identification number: model_id: bc2574197c72a3236c54c8600f455aea. We selected this model, which uses PDB code entry 2vz9 as a template [61], since its NB3F18 contacting portion, comprising the residues 2326–2451, were located at the model’s surface. Consequently, only these residues were taken to build the NB3F18-FREM2 complex. The selected polypeptide fragment was subjected to a short optimization/relaxation run with CHARMM [59], and the structure was checked by the WHAIF modeling suite [62]. Subsequently, both 3D models were submitted to the protein–protein docking suite implemented as the Rosetta online server (http://rosie.rosettacommons.org, accessed on 5 February 2020) [63]. The initial position of the interacting proteins was generated on the basis of PROBIS analysis of protein binding sites [64], revealing the N-terminal surface of NB3F18 and peripheral domain residues F2371-A2393 in FREM2 as the principal interaction surfaces. To check the stability and correctness of the complex, the best resulting complex structure was put into a cube of 22,908 water molecules together with 69 sodium and 65 chloride ions for electrical neutrality and subjected to 150 relaxation steps (50 steps of s.d. optimization, 50 steps of optimization by the adopted basis Newton–Raphson method, and 50 steps of descent lattice optimization) followed by 1000 ns of constant pressure and temperature (CPT) dynamic simulation (300 K, 1 bar, and time step of 1 fs) invoking the EWALD summation for calculating electrostatic interactions [59,60].

### 2.8. Modeling the 3D Interaction Between Nanobody NB3F18 and Human FREM2 with AlphaFold3.0

We modeled the entire FREM2 protein sequence (Q5SZK8-1) and simulated its interactions with NB3F18 using AlphaFold3.0 [65] and analyzed it with GitHub AF3_Results_Visualization.ipynb [66]. To obtain better prediction results, we analyzed the sequence of the recombinant protein fragment of human FREM2 used in our experiments and added its neighboring regions—calix-beta 5 upstream and a disordered region downstream of the Q5SZK8-1 sequence (819 amino acids). To further assess the binding between the top two models, we used an additional docking software, HDOCK version 1.1 [67].

### 2.9. NB3F18 Labeling with Alexa Fluor 546 Dye and SDS-PAGE Analysis

NB3F18 was labeled using a modified protocol of Ries et al. [68]. Briefly, NB3F18 (1 mg/mL) in 0.2 NaHCO_3_ (pH 8.2) was incubated with Alexa Fluor 546 (Molecular Probes Alexa Fluor 546 Protein Labelling Kit, Life Technologies, Carlsbad, CA, USA) at 1:1.5 and 1:5 molar ratios and incubated in the dark at room temperature with constant stirring. To remove unreacted dye, the buffer was exchanged for phosphate-buffered saline (PBS) on a 7 kDa molecular mass cut-off Zeba Spin Desalting Column (Thermo Fisher Scientific), and the labeled NB3F18-^546^Alexa was stored at 4 °C until further use.

NB3F18-^546^Alexa was analyzed by sodium dodecyl sulphate polyacrylamide gel electrophoresis (SDS-PAGE, 12.5% acrylamide gel) [69] under reducing conditions (0.5% (*m*/*v*) SDS, 10% (*v*/*v*) glycerol, 5 m MDTT, and 30 mM Tris–HCl, pH 6.8). Gels were imaged on a ChemiDoc MP System gel imager (Bio-Rad, Hercules, CA, USA). The fluorescence of Alexa Fluor 546 was detected using a Green Epi illuminator and a 605/50 emission filter using the ChemiDoc MP Alexa 546 application.

### 2.10. Flow Cytometry

Cells were treated with 1 µM NB3F18-^546^Alexa for 24 h. Cells were harvested by centrifugation at 300× *g* for 10 min and resuspended in 250 µL Hank’s Balanced Salt Solution (HBSS). Flow cytometry analysis was performed using a FACSCalibur system equipped with a 488 nm Argon-ion laser using the CellQuest software (Becton Dickinson, Franklin Lakes, NJ, USA). The Alexa Fluor 546 fluorescent signal was collected using a FL-3 (650LP) filter. At least 2 × 10^4^ events were analyzed per sample.

For quantification of anti-FREM2 antibody binding on the surface receptor of NCH421K, NCH644, U87MG, U251MG, and astrocytes, we resuspended 100,000 cells in PBS with 5% bovine serum albumin (BSA). The cells were centrifuged at 300× *g* for 10 min, and the pellet was resuspended in PBS with 5% BSA with anti-FREM2 antibody (SAB3500517 produced in rabbit; Sigma-Aldrich) at a 1:30 dilution. After a 60 min incubation at 4 °C in dark, cells were centrifuged at 300× *g* for 10 min, and the supernatant was discarded. The cells were then washed twice and resuspended in PBS with 5% BSA containing the secondary goat anti-rabbit Alexa Fluor 488 antibody for 30 min at 4 °C in the dark. Finally, the cells were analyzed by flow cytometry on a FACSCalibur system equipped with a 488 nm Ar-ion laser using CellQuest software (Becton Dickinson) and the FL-1 (530/30) filter. We used unstained cells and cells stained only with the secondary antibody as controls. At least 2 × 10^4^ events were analyzed per sample.

### 2.11. Immunofluorescence Studies

For astrocytes, coverslips were pre-treated with poly-D-lysine for at least 2 h in a humidified incubator at 37 °C and 5% CO_2_. Approximately 100,000 glioblastoma stem cells/well were seeded in wells of a 12-well plate. Differentiated glioblastoma cells and astrocytes were seeded at approximately 20,000 cells/well and incubated for 24 h. NB3F18-^546^AlexaFluor (1.5 µg/mL) was then added to the wells, and the cells were incubated for an additional 24 h. The cells were incubated with BioTracker™ 490 Green Cytoplasmic Membrane Dye (SCT106, Millipore, Burlington, MA, USA) for 20 min (U251MG, U87MG and astrocytes) or 30 min (NCH644 and NCH421K). Cells were washed twice with warm medium, once with PBS, and finally fixed for 15 min at room temperature with freshly prepared ice-cold 4% formaldehyde. Cells were washed three times with PBS. U251MG, U87MG, and astrocytes were then mounted on glass slides using ProLong^®^ Diamond Antifade Mountant with 4′,6-diamidino-2-phenylindol (DAPI, P36962, Thermo Scientific). After washing with PBS, NCH644 and NCH421K cells were centrifuged for 5 min at 200× *g*. The supernatant was removed completely, and the cells were resuspended in 20 µL PBS. The cell suspension in PBS was transferred to glass slides and left at 37 °C for PBS to evaporate. NCH644 and NCH421K were then mounted on glass slides using ProLong^®^ Diamond Antifade Mountant with DAPI. After the addition of fluorescent agents, experiments were performed in the dark. For each cell line, we prepared 4 slides as described: 1× nanobody and membrane staining dye and three control slides, namely 1× only nanobody, 1× only membrane staining dye, and 1× cells with DAPI.

Microscopy slides were observed under an Axio Observer Z1 LSM 710 (ZEISS, Jena, Germany) inverted confocal laser scanning microscope with a Plan-Apochromat 63/1.40 oil objective. Images were analyzed using ZEN 2010, v. 6.0.0.485 software (ZEISS).

### 2.12. Preparation and Characterization of NB3F18-Au^5nm^ Conjugates

NB3F18 was conjugated with 5 nm gold particles using a 5 nm Ni-NTA-Nanogold kit (2082, Nanoprobes, Yaphank, NY, USA) using a modified protocol designed for labeling His-tagged protein complexes in solution [70]. PBS in which NB3F18 (1 mg/mL) was diluted was changed to 20 mM Tris (pH 7.6) with 150 mM NaCl (TBS). In order to avoid harsh concentration changes that may result in NB3F18 aggregation, we used an Amicon Ultra-0.5 mL Centrifugal Filter Device (UFC5010BK, Merck, NJ, USA) with a 10,000 molecular mass cut off and changed the buffer in multiple steps [71]. With a series of 21 centrifugations at 5500× *g* each for 1 min, we achieved over 99% buffer replacement. The NB3F18 concentration was adjusted to 1 µM and combined with 5 nm gold particles in suspension in a 1:1.5 ratio. Conjugates were incubated for 30 min on ice. To remove unbound NB3F18 and replace the buffer with TBS, conjugates were centrifuged at 4000× *g* for 1 min using Ultracel YM-100 membranes (Amicon, Millipore). After 7 centrifugations, 99% of the unbound NB3F18 was removed. A NanoDrop ND-1000 spectrophotometer (Thermo Fisher Scientific) confirmed negligible protein elution in the collected elution buffer, thereby validating effective conjugation. We adjusted the buffer volume to reach 1 mg/mL. The NB3F18-Au^5nm^ conjugates were stored at 4 °C until further use.

The NB3F18-Au^5nm^ conjugates were analyzed by negative staining. The conjugate suspension was diluted tenfold with water and adsorbed onto carbon-coated Cu grids. Grids were washed with H_2_O and contrasted with 2% uranyl acetate (22400, Electron Microscopy Sciences, Fort Washington, PA, USA) for 6 min at room temperature. The excess sample was blotted off, and the grids were allowed to dry on air before examination with a Philips CM100 transmission electron microscope (Philips, Amsterdam, The Netherlands).

### 2.13. Transmission Electron Microscopy Analysis

U87MG and U251MG (2.64 × 10^5^ cells/well) cells were seeded into a sterile 6-well dish and incubated until they reached close to 100% confluence. The bottom of a 6-well plate was pre-treated with poly-D-lysine before seeding astrocytes (1 × 10^5^ cells/well) and incubation as described above. NCH644 and NCH421K (5 × 10^5^ cells/well) were seeded into a sterile 6-well plate and incubated for 24 h. All cells were incubated with 100 mg/mL NB3F18-Au^5nm^ for 2 h. Control U251MG cells were not incubated with gold particles. After incubation, cultures were washed with medium without NB3F18-Au^5nm^ and fixed with 2.5% glutaraldehyde in 0.1 M cacodylate buffer, pH 7.2, for 1 h at room temperature. Cells were washed with 0.33 M sucrose in 0.1 M cacodylate buffer 3 times for 10 min at room temperature. Cells were post-fixed with 1% OsO_4_ and 0.8% K-ferrocyanide in 0.1 M cacodylate buffer for 30 min at room temperature in the dark. Cells were then washed with water and incubated with 2% uranyl acetate for 30 min at room temperature in the dark. At last, cells were dehydrated in a rising concentration of EtOH and embedded in Epon 812 (Electron Microscopy Sciences). Then, 60 nm ultrathin sections were cut using the Leica Ultra Cut device (Leica Microsystems, Vienna, Austria) and were analyzed with transmission electron microscopy (TEM).

### 2.14. Cell Proliferation Assay

Flat bottom wells of 96-well plates were pre-treated with poly-D-lysine. In a total volume of 100 µL, 4000 cells/well were seeded in triplicates. Cells were incubated for 2 h, and then 10 µg/mL and 100 µg/mL nanobody NB3F18 was added. Cells were incubated for 24 h, 48 h, and 72 h. After these incubations, 10 µL 2-(4-iodophenyl)-3-(4-nitrophenyl)-5-(2.4-disulfophenyl)-2H-tetrazolium, sodium salt—WST-1 reagent (05015944001, Roche, Basel, Switzerland) was added to each well. Cells were incubated for 2 h. Absorbance was measured at 450 nm and 620 nm using a Synergy H4 Hybrid microtiter plate reader (BioTek, Shoreline, VT, USA). Controls were treated the same as the rest of the cells but without the addition of nanobodies. Experiments were repeated three times.

Cell viability (survival) was determined as follows: for each sample, we first determined the ratio between A_450nm_ and A_690nm_. Then, the blank (cell medium) was subtracted from each value. The percentage of cell survival was determined as the ratio between treated and non-treated (control) cells multiplied by 100.

### 2.15. Statistical Analysis

Data were statistically analyzed in GraphPad Prism 6 (GraphPad Software Inc., San Diego, CA, USA). For flow cytometry, samples were analyzed using two-way analysis of variance (ANOVA) with Bonferroni correction for multiple comparisons. For the cell proliferation assay, samples were analyzed using one-way ANOVA and Holm–Sidak’s corrections for multiple comparisons. In all cases, *p* ≤ 0.05 was considered statistically significant (* *p* ≤ 0.05; ** *p* < 0.01; *** *p* < 0.001; **** *p* < 0.0001).

## 3. Results

### 3.1. Identification of an Anti-FREM2 Nanobody and Its Preparation for In Vitro Use

A suitable polypeptide was chosen to select and identify an anti-FREM2 nanobody from the existing library. We computed the parameters of a commercially available recombinant protein fragment (polypeptide) of human FREM2 used for library enrichment, screening, and SPR with the ProtParam tool [54]. This FREM2 protein fragment contains 126 amino acids, has a molecular mass of 13,997.95 Da, and has a theoretical isoelectric point (pI) of 4.57. According to the UniProt protein database [72] which uses the code Q5SZK8 for FREM2 [73], the FREM2 peptide that we used in our studies belongs to the extracellular part of the FREM2 protein, starting at amino acid residue 2325 and ending at amino acid residue 2451 (section Subcellular location, subsection Topology, and section Sequences (2), Isoform 1).

The process of nanobody preparation is schematically presented in Figure 1. For nanobody selection, a previously constructed anti-glioblastoma immune library [49] was expressed on phage to retrieve binders to the FREM2 peptide in three rounds of biopanning. Periplasmic protein extracts of potential nanobody clones (192 bacterial clones after the 2nd and 192 bacterial clones after the 3rd round of biopanning) were screened for specific recognition of the FREM2 peptide via ELISA. Periplasmic extracts of clones showing a 10× higher signal in wells coated with FREM2 peptide compared to wells without peptide were selected as potential binders (5 nanobody clones after 2nd and 61 nanobody clones after 3rd round of biopanning). After 60 min, our selected nanobody showed a 60× higher signal in the well coated with FREM2 polypeptide compared to the well without such coating (absorbances of 3.587 and 0.091, respectively, Appendix A), suggesting specificity of the selected nanobody for the FREM2 peptide.

Of the 66 bacterial clones that were selected with ELISA, 62 contained fragments of the appropriate size (~600 bp) and were sequenced. From the potential clones, 16 nanobodies had identical sequences, and 44 had truncated sequences. Only two nanobodies, NB2F111 and NB3F18, had the complete nanobody sequence with the characteristic amino acid sequences at the amino-terminus (i.e., QVQL) and carboxy-terminus (i.e., TVSS), as well as a distinct CDR1, CDR2, and CDR3. On a first SPR test, only NB3F18 showed binding to the FREM2 peptide and was chosen for further analysis. For NB3F18, amino acids at FR2 at ImMunoGeneTics (IMGT) numbering position 42, 50, and 52 (F, R, and L, respectively) corresponded to variable domains of the heavy chain of heavy-chain antibodies’ (VHH) hallmarks. This nanobody has an unusually long CDR2 and a short CDR3 sequence (Figure 2). The nanobody sequence was analyzed with the ProtParam tool [54], and the information is presented in Table 1.

The expressed nanobody was purified (Appendix A), and specific expression was confirmed with Coomassie-stained SDS-PAGE (Appendix A).

### 3.2. NB3F18 Binds to the Recombinant Protein Fragment of Human FREM2 with Medium Affinity

We used SPR to study the interaction of NB3F18 with the chip-immobilized FREM2 peptide. Vimentin was used as a negative control. Results revealed fast association and dissociation kinetics of the NB3F18-FREM2 complex (Figure 3A) and an equilibrium dissociation constant (K*_D_*) of 1.140 µM ± 0.025 µM (Figure 3B). We measured an insignificant interaction between NB3F18 and vimentin (Figure 3C). Results show that the NB3F18-FREM2 complex has intermediate affinity in vitro.

### 3.3. Three-Dimensional Modeling Implies Stability of the NB3F18-FREM2 Complex for 1000 ns

The visualization of the putative complex between NB3F18 and a 126-residue long portion of FREM2 (residues F2371-A2393) was constructed using homology-built models of both components: the Swiss-Prot modeler was employed to build a three-dimensional (3D) model of the NB3F18 nanobody, using a very high identity template (63%; PDB code: 5o0w). The 3D model structure of FREM2 was found in the Modeller’s database of Sali’s lab. From the latter, only a portion was taken, and, subsequently, both structures were submitted to the Rosetta WEBB suite that suggested several putative protein–protein interaction patterns. Among them, the one with the highest score was chosen and further processed locally, using CHARMM, a 3D modeling and simulation program. After several steps of relaxation, optimization, and equilibration, the complex was subjected to an extensive in silico dynamic simulation. The result shown in Figure 4 shows the complex between NB3F18 and a possible recognized epitope of FREM2. Superimposed (depicted) are the first (nanobody—blue; FREM2—turquoise) and final (nanobody—red; FREM2—purple) frames after 1000 ns of a molecular dynamic run.

Figure 4 and Figure 5 show that the molecules remain in close contact throughout the entire time period. Figure 5, which shows a continuous change in RMS during the 1000 ns simulation, shows a suggested significant increase of approximately 15 Å, but this is a consequence of the relatively unstable FREM2 fragment. Such a conformational change is to be expected for a short polypeptide derived from a large protein. Despite this, the RMS of the complex became stable after 250 ns (Figure 5), in parallel with the formation of three electrostatic interactions: K26-E2335, K32-E2362, and V2(N-terminal)-D2384 between NB3F18 and FREM2, respectively. The conformation of NB3F18 did not change significantly throughout the entire simulation run, as shown by the super-positions in Figure 4. Although the two proteins remained in close contact throughout the simulation run, it was found that the contacting portion of FREM2 accommodated the conformation quite significantly. Nevertheless, the three electrostatic interactions formed a strong anchor yielding a significant binding energy.

Based on the identified NB3F18 interaction with 126 amino-acid residues (F2371-A2393) of FREM2, we modeled their complex, which appeared stable throughout a 1000 ns molecular dynamics simulation run. This result corroborated the affinity determined by SPR. A relatively long, 1000 ns in silico molecular dynamic simulation experiment suggested a possible stable interaction between NB3F18 and the 126-amino-acid-residue-long FREM2 fragment.

### 3.4. AlphaFold3.0 Shows Low Prediction Accuracy for FREM2 Protein and in Simulating Its Interactions with NB3F18

Modeling the FREM2 protein and simulating its interactions with NB3F18 using AlphaFold3.0 was realized with limited success. As seen in Figure 6, the predicted score for NB3F18 was relatively high, with a predicted template modeling score (pTM) of 0.86 (Figure 6A,B.4). However, we were unable to obtain a reliable prediction for the entire FREM2 protein sequence (Q5SZK8-1) which had a pTM below 0.4 (Figure 6A,B.1). Modeling the recombinant FREM2 protein fragment only and in combination with its neighboring regions calix-beta 5 upstream and a disordered region downstream of the Q5SZK8-1 sequence (819 amino acids) resulted in an improved pTM of 0.53 (Figure 6A,B.2) and 0.58 (Figure 6A,B.3), respectively. After aligning the best-predicted models from AlphaFold3.0, we observed a low binding prediction between the recombinant FREM2 peptide and its neighboring regions and NB3F18 (pointed to with an arrow) with an interaction prediction score (ipTM) of 0.13 and overall low accuracy of the prediction (0.45) (Figure 6A,C.1). plDDT scores for FREM2 with its neighboring regions and NB3F18 following binding are given in Figure 6C.2 and Figure 6C.3, respectively, and indicate a low modeling score.

The HDOCK docking software yielded the top three calculated docking scores of −304.32, −247.79, and −246.31 (Figure 6D), suggesting a high likelihood of correct positioning between the two proteins. However, the ligand root mean square deviation (RMSD) values were around 50 Å, indicating poor binding affinity.

### 3.5. NB3F18 Labeling with Alexa Fluor 546 Dye

NB3F18 was successfully labeled with Alexa Fluor 546 dye at a 1:1.5 molar ratio, which provided optimal signal intensity and antigen binding properties. Higher concentration of Alexa Fluor 546 dye in the labeling procedure resulted in fluorescent conjugates of NB3F18 that readily formed aggregates and were unsuitable for subsequent applications. The initial concentration of NB3F18-546Alexa was 2.69 mg/mL (368 μg total). After desalting, the concentration of NB3F18-546Alexa was 1.5 mg/mL, corresponding to a total recovery of 210 μg labeled nanobody. The calculated dye-to-protein ratio indicated an average of 1 mole of conjugated dye per mole of NB3F18. No specific assays to assess the functional integrity of NB3F18-546Alexa post-labeling were performed in this study. The success of labeling was demonstrated by SDS-PAGE analysis, followed by the detection of Alexa 546 dye on a fluorescence gel imager. This confirmed that NB3F18-^546^Alexa did not contain free Alexa 546 dye (Appendix A).

### 3.6. Cellular Specificity of NB3F18 Towards Glioblastoma Stem Cells NCH644

Cellular specificity was quantified by flow cytometry. The use of controls allowed us to determine the baseline autofluorescence of the cells and remove it from the analysis. As presented in Figure 7A, NB3F18 binds 65.36% and 36.47% of the gated glioblastoma stem cells NCH644 and NCH421K, respectively. It also binds 19.04% of the U251MG, 25.25% of the U87MG, and 25.35% of the astrocytes (Figure 7A). This suggests that NB3F18 binds to glioblastoma stem cells to a higher extent than to differentiated glioblastoma cells. For comparative analysis, we tested the binding specificity of an anti-FREM2 antibody which showed similar results (Figure 7B). The anti-FREM2 antibody binds to 58.79% of the NCH644, 65.82% of the NCH421K, 13.10% of the U251MG, 20.35% of the U87MG, and 39.47% of the astrocytes. Results of the biological replicates are shown in Table 2, and the statistical analysis is graphically presented in Figure 7C.

### 3.7. NB3F18-^546^Alexa Localizes to the Surface of Glioblastoma Stem Cells NCH644

To examine the cellular localization of FREM2 in glioblastoma cells and astrocytes, we performed live cell imaging that allows for incubation of NB3F18-^546^Alexa with the cells while they were still in a metabolically active state. This enabled the nanobody to enter the cell and bind to its intracellular target. The cellular membrane was detected with a commercially available membrane staining dye (green). Figure 8 shows that besides binding intracellularly, NB3F18 (red) bound to the surface of glioblastoma stem cells NCH644 and NCH421K (white arrows), while its localization in the differentiated glioblastoma cells U251MG and U87MG and in astrocytes was cytoplasmic.

Furthermore, NB3F18 localized differently in differentiated glioblastoma cells (Figure 8). While the nanobody signal in U87MG was distributed throughout the cytoplasm, the localization of NB3F18 in U251MG remained perinuclear. Localization in astrocytes was cytoplasmic without any apparent specificities. Since FREM2 was originally reported to be found on the basement membrane, our results suggest that it may also appear with altered localizations.

### 3.8. NB3F18 Conjugation with Gold Nanoparticles

On negative stained grids, we noted individual electron dense granules that measured approximately 5 nm in diameter, confirming that the prepared conjugates did not aggregate (Appendix A).

### 3.9. Subcellular Localization of NB3F18-Au^5nm^ in Glioblastoma Cells and Astrocytes

To study the binding of NB3F18 at higher resolution, we incubated cells with NB3F18-Au^5nm^ and observed the cell in TEM. NB3F18-Au^5nm^ was detected on the plasma membrane of NCH644, NCH421K, U87MG, and U251MG cells. In addition, we observed NB3F18-Au^5nm^ in endocytic vesicles, early endosomes, and multivesicular bodies, with the later representing late endosomes (Figure 9A–D), as well as in lysosomes (Figure 9B). Some conjugates were also bound to the plasma membrane of astrocytes (Figure 9E), but no signal was detected inside the cells. Invaginations of the plasma membrane show the early stages of NB3F18-Au^5nm^ endocytosis (Figure 9G,H). No coats were detected on the cytosolic face of the membrane, suggesting a clathrin-independent mechanism of NB3F18-Au^5nm^ internalization. Additionally, the diameters of endocytic vesicles were 150–500 nm, which is in the range of macropinosomes’ size (Figure 9C,D). We did not detect gold particles on or inside the control cells (Figure 9F).

### 3.10. NB3F18 Is Cytotoxic Only to Glioblastoma Stem Cells NCH644

Next, we tested whether NB3F18 has a cytotoxic effect for glioblastoma cells using a WST-1 colorimetric assay. We determined the cytotoxicity of 10 µg/mL and 100 µg/mL NB3F18 on glioblastoma cells and astrocytes over three different time periods: 24 h, 48 h, and 72 h (Figure 10). The greatest negative effect was observed on the cell viability of NCH644 after 48 h and 72 h of incubation when using 100 µg/mL NB3F18. Cell viability was reduced to 84.7% and 55.46%, respectively. Incubation with 100 µg/mL NB3F18 reduced the cell viability of NCH421K to 76.41%. NB3F18 affected the growth of the other studied cell lines to a much lesser extent (Table 3). Importantly, NB318 did not drastically affect the viability of astrocytes even after prolonged incubation.

## 4. Discussion

To obtain more information about the distribution of FREM2 in different types of cells in tissues of the human body [74], we analyzed the *FREM2* expression levels using data available from The Human Protein Atlas [75] which provides combined transcriptomics and antibody-based proteomics data aiming to map human proteins at a single-cell and spatial resolution. As it can be seen in Figure 11, at the single-cell RNA level, *FREM2* shows specificity for astrocytes (glial cells, 43.7 nTPM), glandular and luminal cells (42.3 nTPM), and skeletal myocytes (45.9 nTPM).

When it comes to single nuclei cluster types, *FREM2* shows specificity for Bergmann glia (144 nTPM), LAMP5-LHX6, and Chandelier neuronal cells (73.5 nTPM) as presented in Figure 12.

*FREM2* RNA expression levels were analyzed in different TCGA and experimental validation cohorts (Figure 13) and showed that *FREM2* has low overall cancer specificity [76]. However, among the analyzed samples, the glioblastoma validation cohort showed the highest *FREM2* expression of 8.5 TPM in the 58 analyzed glioblastoma samples followed by 4 TPM in the 50 samples included in the breast invasive carcinoma validation cohort. In both cases, the *FREM2* RNA expression level in TCGA cohorts was lower, i.e., 1.7 TPM for glioblastoma and 0.1 TPM for breast invasive carcinoma compared to the *FREM2* RNA expression level in the corresponding experimental validation cohorts.

Images from The Human Protein Atlas provide a comprehensive map of the *FREM2* expression pattern across various physiological and pathological conditions. The results from Figure 11, Figure 12 and Figure 13 indicate that FREM2 has a cell-specific expression pattern which may be a result of a tissue-specific role. The higher *FREM2* expression in astrocytes suggests that NB3F18 can be used for specifically targeting GSCs while minimizing side-effects on non-neoplastic cells. At last, the high *FREM2* expression in glioblastoma cohorts strengthens the potential use of FREM2 as a therapeutic target.

In addition, we constructed the FREM protein network (Figure 14) using the publicly available program STRING [77,78,79,80,81,82,83,84,85,86,87,88,89], with the inclusion of no more than 50 interactors in the first shell and a confidence value of 0.4. Functional enrichment (Kyoto Encyclopedia of Genes and Genomes (KEGG) pathways) analysis showed that FREM2 is related to extracellular matrix (ECM)–receptor interactions (violet), the ErbB signaling pathway (yellow), focal adhesion (green), and the phosphoinositide 3-kinase (PI3K)/Akt signaling pathway (red). Nevertheless, the majority of interactors are not assigned into any of the most significant KEGG pathways, indicating that the role of FREM2 is largely unknown.

Here, we report the isolation, characterization, and functional validation of a nanobody, NB3F18, directed against the corresponding protein of *FREM2*. Due to the membrane-associated localization of FREM2 and its overexpression in glioblastomas, the FREM2-specific nanobody can be used for selective targeting of glioblastoma cells with minimal damage to adjacent cells and without the need for entering the cell. Hence, it was important that the nanobody interacted with the extracellular part of FREM2. With flow cytometry, we showed that NB3F18 binds to NCH644 glioblastoma stem cells to a greater extent than to the other analyzed cells. The binding pattern of the positive control, the anti-FREM2 antibody, is similar to the one of NB3F18. This result validates the experimental setup, enhances reliability, and confirms our findings that NB3F18 binds to FREM2 and is more specific towards GSCs. The stemness of GSCs NCH644 and NCH421K was proven in our previous publication [90]. As shown in the cell proliferation assay, even though NB3F18 binds to astrocytes, it does not significantly affect their viability even after extended exposure to NB3F18 with minimal cytotoxic effects at high concentrations (100 µg/mL). This implicates that NB3F18 has limited off-target effects on non-tumor cells, i.e., astrocytes. Moreover, as revealed by the immunofluorescence results, NB3F18 showed specificity for the cellular membrane of NCH644 and NCH421K glioblastoma stem cells, which was not observed for the other investigated cells. FREM2 is known to be localized in the cytosolic portion of U251MG cells [91,92]. Nonetheless, during embryogenesis in mice, Fras1, Frem1, and Frem2 are predominantly found in basement membranes and remain in small quantities in adult basement membranes [92], which we confirmed in our immunofluorescence experiment. The expression of the Frem2 transcript is restricted both in space and time, and it emerges prior to cellular reorganization events. Therefore, the expression of Frem2 could potentially modify the ECM in a dynamic manner to establish a foundation for cell migration and rearrangements during embryonic development [93]. As can be seen in Figure 6, NB3F18 presents with a different cytoplasmic localization in U87MG and U251MG glioblastoma cells. This may be a result of their different growth dynamics, cytokine expression, and morphology [94]. Using TEM, we were able to localize NB3F18-Au^5nm^ at the ultrastructural level. Compared to immunofluorescence, the limitation of the TEM experiment was the short incubation time (24 h versus 2 h, respectively) because of the potential cytotoxicity of the gold particles [95]. Namely, at longer incubation times, e.g., 24 h, gold particles do not only cause cytotoxicity but also ultrastructural changes [96]. NB3F18-Au^5nm^ were observed both at the cellular surface (plasma membrane) and inside the cells. Despite the difference in incubation times, the results from the immunofluorescence and TEM experiments for NCH644 and NCH421K are comparable. Since upon longer incubation NB3F18-^546^Alexa was detected inside U87MG, U251MG, and astrocytes, detection of NB3F18-Au^5nm^ at the plasma membrane of these cells may also represent the beginning of the internalization [96].

Several mechanisms targeting glioblastoma stem cells have been tried [16,97,98,99] but so far without success. Due to their natural origin and biocompatibility, nanobodies are generally non-toxic and suitable for use in human subjects. The observed cytotoxicity in glioblastoma stem cells may be due to the functional modulation of FREM2 by NB3F18. As supported by our STRING analysis, FREM2 plays a crucial role in regulating the differentiation and migration of glioblastoma stem cells through ECM and integrins. However, the exact roles of FREM2 are currently unknown and so is its mechanism of action. Timmer et al. reported that Frem2 is expressed primarily during embryogenesis [93]. On the other hand, cancer stem cells are known to have embryonic stem cell signatures [100] which may be the reason for the different affinity and cytotoxicity.

Even though AlphaFold3.0 is a powerful tool for structural modeling, it presented with limitations in predicting complex protein interactions such as the ones between FREM2 and NB3F18 based on the low confidence scores. On the other hand, molecular dynamics simulations and in silico models built with CHARMM and Rosetta docking validated the interaction and revealed key electrostatic interactions contributing to the stability of the complex. The low prediction accuracy of AlphaFold3.0 suggests that the protein structural complexity and conformational dynamics are not always fully captured by current computational models. Therefore, computations tools can complement but not replace experimental validation (e.g., SPR) especially for complex proteins such as FREM2. As shown by SPR, the affinity of NB3F18 for FREM2 is moderate, i.e., in the micromolar range. Antibodies with moderate affinity are considered as better vehicles for the delivery of different agents than high-affinity antibodies [101]. As observed by Adams et al., this is because high affinity limits tumor penetration and intratumoral diffusion as antibodies are trapped at the tumor periphery. Therefore, the ability of NB3F18 to bind to glioblastoma stem cells to a greater extent than to the rest of the analyzed cells and the formation of a moderate affinity complex between NB3F18 and FREM2 are consistent with the purpose of our research to develop a molecular tool able to target a specific subset of glioblastoma cells. From the 3D binding modeling, we conclude that the interaction is sufficiently stable, and the nanobody can be used not only for targeting purposes but also as a drug carrier to avoid systemic toxicity and to deliver large amounts of drugs more successfully [102]. Moreover, such nanobody–antigen affinity as well as fast association and dissociation kinetics are desired characteristics for techniques such as Peptide-PAINT [103], i.e., for single-molecule detection at high resolutions. Because of their small size and high affinity, nanobodies may become utile for labeling regions that are currently inaccessible for antibodies. Also, their use in Peptide-PAINT can be very beneficial for the antigens that are not nicely visualized using currently available probes such as monoclonal and polyclonal antibodies because of suboptimal binding efficiency.

Some of the problems in the clinical management of glioblastomas can be circumvented using nanobodies. Due to their small size and higher content of exposed hydrophilic amino acids, they might cross the plasma membrane which for NB3F18 was also shown by our TEM analysis. Some of them have also been observed to pass the blood–brain barrier (BBB) directly (by receptor- and adsorptive-mediated endocytosis) or indirectly (via specialized liposomes or tumor-modified BBB regions) [104]. The high pI of NB3F18 is in the range predicted to pass the BBB according to Li et al. [105]. This is relevant for glioblastomas [106], as it allows them to better penetrate the tumor and gives them the opportunity to reach and access hidden epitopes and target traditionally inaccessible intracellular tumor markers. Nanobodies also have a high degree of modularity that offers greater paratope diversity than traditionally used monoclonal antibodies. This improves target specificity and widens the pool of potential new therapeutic targets [107].

## 5. Conclusions

Nanobodies allow real-time tracing and visualization of proteins as well as modulation of their function. Data obtained from our experiments suggested that NB3F18 can be used as a molecular tool to study and uncover the role of FREM2 in glioblastoma pathogenesis. Based on its specificity for glioblastoma cells, we conclude that NB3F18 can be used in the design of targeted therapeutic approaches, or as a vehicle for targeted delivery of biological or pharmacological agents. Because the nanobody binds to astrocytes but does not affect their viability and growth, we expect there will be fewer off-side effects if administered in vivo. This strengthens the potential of NB3F18 for specific targeting of glioblastoma cells while minimizing adverse effects on healthy brain tissue. In our study, we used commercially available glioblastoma cells and astrocytes because of their robustness. In the future, we plan to validate results in patient-derived glioblastoma cells and in more complex models such as glioblastoma spheroids, organoids, or in vivo.

## Figures and Tables

**Figure 1 antibodies-14-00008-f001:**
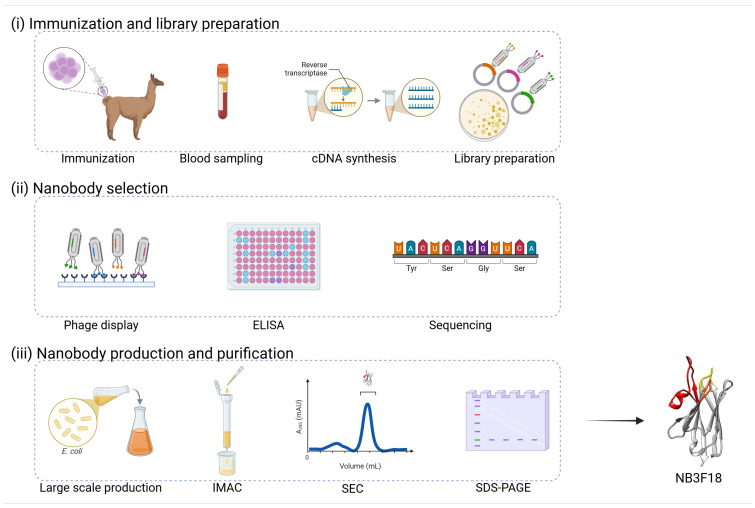
Schematic presentation of nanobody preparation. There are three main phases: (**i**) immunization and library preparation consisting of llama immunization, drawing blood, extraction of mRNA and cDNA synthesis, amplification of the *VHH* genes, insertion into phage display vectors, and library creation; (**ii**) nanobody selection consisting of library enrichment using phage display, i.e., biopanning, nanobody selection with ELISA, and sequencing to identify the nanobodies with complete sequences; and (**iii**) nanobody production and purification consisting of large-scale production of selected nanobodies in *E. coli* host cells, purification with IMAC and SEC, and validation of the purified nanobodies with SDS-PAGE. Image was created with BioRender.com.

**Figure 2 antibodies-14-00008-f002:**
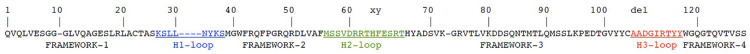
Amino acid sequence of NB3F18. Framework regions 1, 2, 3, and 4 are shown in black. Hypervariable loops H1-, H2-, and H3-loop are underlined and given in blue, green, and red, respectively. The amino acids of the CDR3 (i.e., H3) are in alphabetical order. Numbering is according to the IMGT information system. ‘xy’ was included in the numbering as the H2-loop is longer than usual due to 2 amino acid insertions, while the H3-loop is relatively shorter than most camelid H3-loops as indicated by ‘del’.

**Figure 3 antibodies-14-00008-f003:**
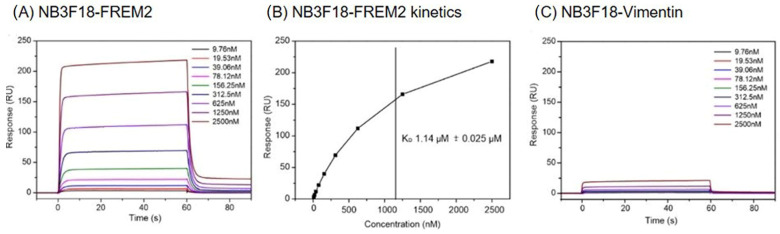
SPR sensorgrams of the NB3F18 nanobody interaction with the immobilized (**A**) FREM2 (990 response units, RU) shown along with the apparent equilibrium dissociation constant (K*_D_*) or (**B**) determined from the response curves as a function of the NB3F18 concentration injected across FREM2. K*_D_* is determined as the mean ± standard deviation of three titrations. (**C**) SPR analysis of the NB3F18 interaction with the immobilized vimentin. NB3F18 was flowed over immobilized polyproteins in serial dilutions as marked on the graphs for 60 s at a rate of 30 µL, and dissociation was followed for 60 s.

**Figure 4 antibodies-14-00008-f004:**
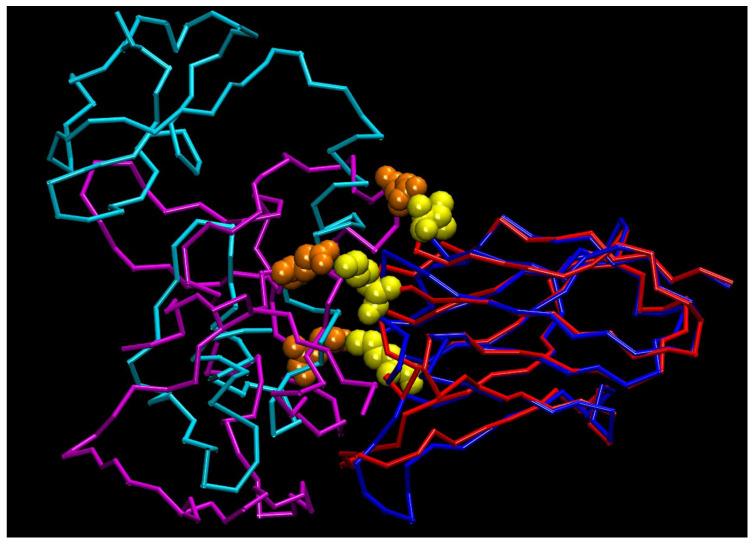
3D model of the complex between nanobody NB3F18 from *Vicuña pacos* and its target human FREM2. The figure represents the starting (NB3F18—blue; FREM2—turquoise) and final frame (NB3F18—red; FREM2—purple) after 1000 ns of molecular dynamic simulation (for details, see text). Note the electrostatic interactions between participating residues N-terV2, K26, K32 (yellow, NB3F18), and D2384, E2335, and E2362 (orange, FREM2) from the uppermost portion down, respectively.

**Figure 5 antibodies-14-00008-f005:**
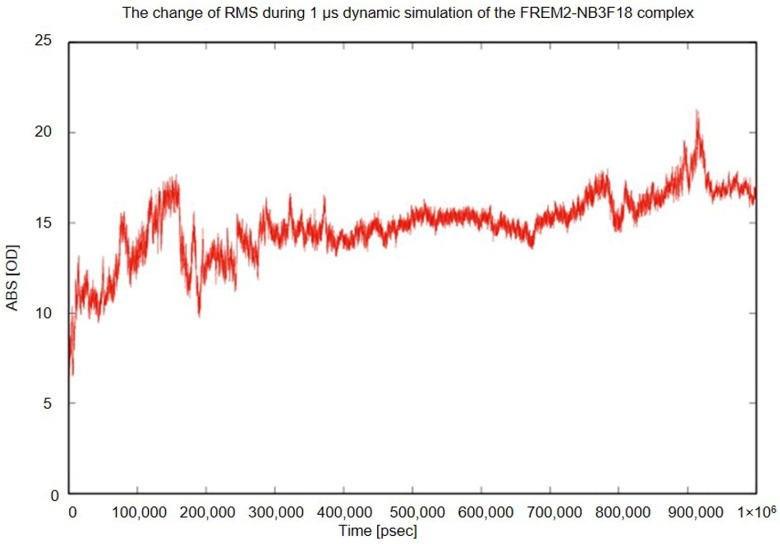
Time-resolved changes in RMS of the complex NB3F18–human FREM2 during 1000 ns molecular dynamic simulation run.

**Figure 6 antibodies-14-00008-f006:**
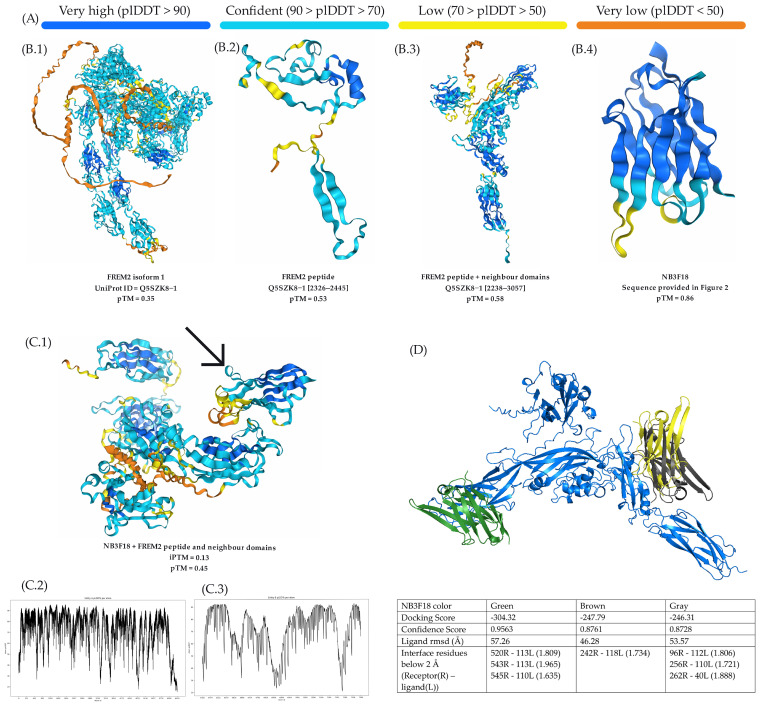
Modeling the FREM2 protein and NB3F18 and their binding predictions using AlphaFold3.0. (**A**) Color scale of the predicted local distance difference (plDDT) used to assess per-residue prediction confidence. This scale applies to the 3D structures shown in **B.1**–**B.4** and **C.1**. (**B**) AlphaFold3.0 predictions of FREM2 and NB3F18 with associated pTM scores. (**B.1**) represents the full-length FREM2 isoform 1 protein sequence obtained from UniProt. (**B.2**,**B.3**) are modeling predictions for subsections of the full-length FREM2 protein shown to bind with NB3F18 in vitro. (**C**) AlphaFold3.0 predictions of interactions between the NB3F18 nanobody and the FREM2 peptide, including neighboring regions with the highest pTM scores. (**C.1**) shows the 3D structure of NB3F18 and the top-predicted FREM2 sequence. The arrow indicates the NB3F18 sequence. (**C.2**) represents the plDDT scores for the FREM2 peptide and neighboring regions following binding with NB3F18. (**C.3**) represents the plDDT scores for the NB3F18 model after binding to the FREM2 peptide and neighboring regions. (**D**) Predicted docking sites between NB3F18 and the FREM2 peptide (including neighboring regions) generated by HDOC. The table below the prediction model shows the docking scores, confidence values, ligand root mean square deviation (RMSD) scores, and interface residue pairs below 2 Å for the top 3 NB3F18 nanobody binding sites.

**Figure 7 antibodies-14-00008-f007:**
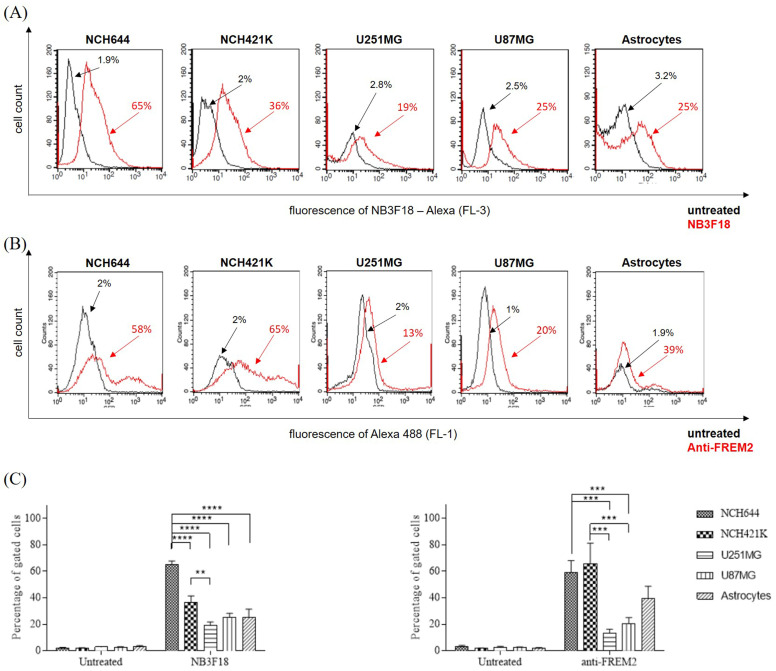
Binding of NB3F18 and anti-FREM2 antibody to cells as analyzed by flow cytometry. (**A**) NB3F18 shows strong binding to glioblastoma stem cells, especially to NCH644, when compared to the other glioblastoma cell lines (U251MG and U87MG) and reference astrocytes. (**B**) anti-FREM2 antibody shows strong binding to glioblastoma stem cells NCH644 and NCH421K when compared to the other glioblastoma cell lines (U251MG and U87MG) and reference astrocytes. (**C**) Statistical analysis of the results adjusted using the Bonferroni correction. NB3F18 binds more strongly to NCH644 glioblastoma stem cells compared to the other analyzed cells. ** *p* < 0.01; *** *p* < 0.001; **** *p* < 0.0001.

**Figure 8 antibodies-14-00008-f008:**
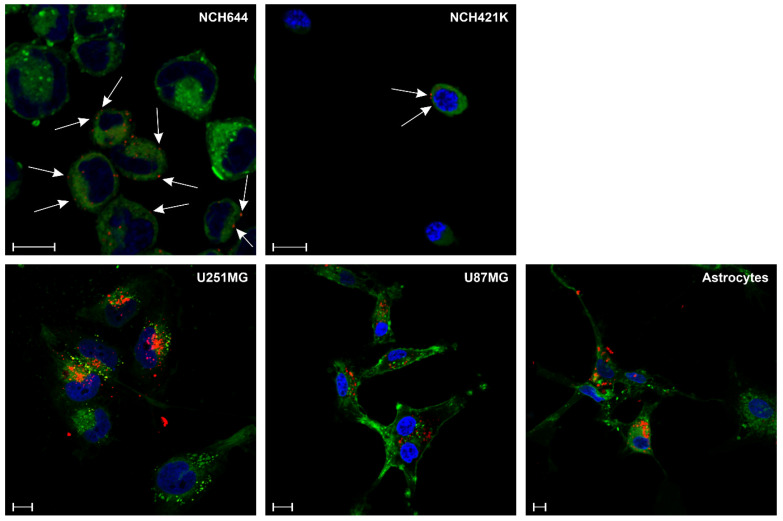
Immunofluorescent analysis of the localization of NB3F18-^546^Alexa on different glioblastoma cells and astrocytes. Red—NB3F18-^546^Alexa; green—membrane-staining dye BioTracker; blue—DAPI. In NCH644 and NCH421K glioblastoma stem cells, the signal from NB3F18-^546^Alexa was seen also on the cell surface (white arrows). This is not observed in the other analyzed cells. Scale bars correspond to 10 μm.

**Figure 9 antibodies-14-00008-f009:**
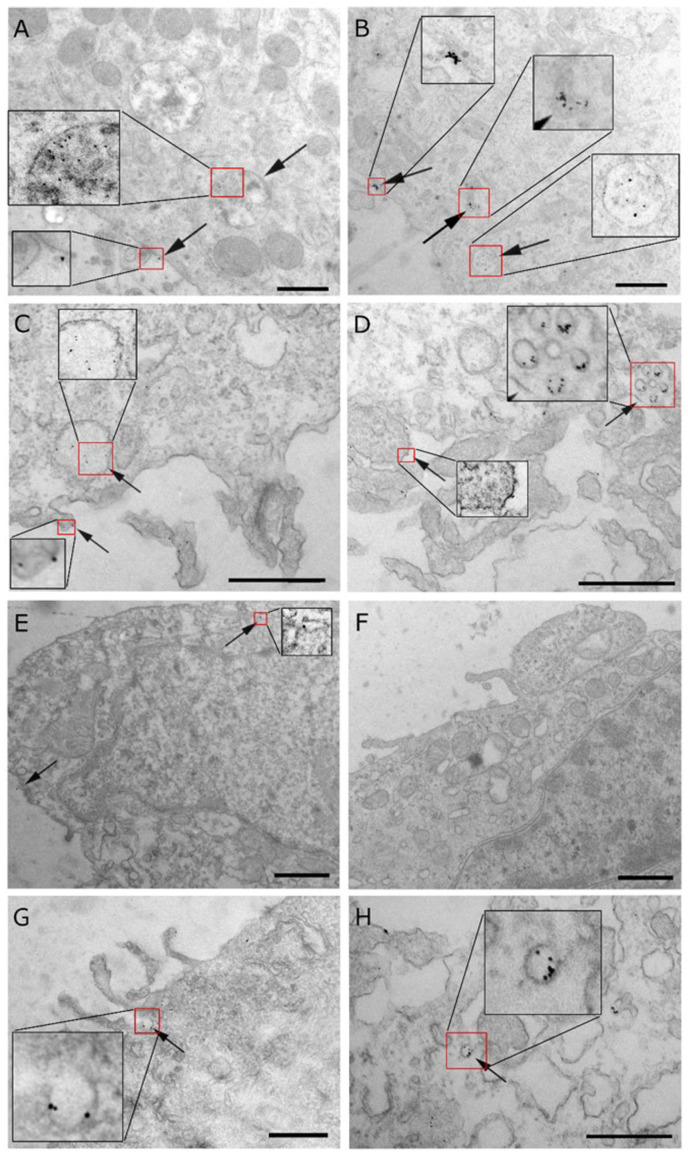
Cellular distribution of NB3F18-Au^5nm^ observed with TEM. (**A**) NCH644 with NB3F18-Au^5nm^ bound to the plasma membrane and inside late endosome/multivesicular body (arrows). (**B**) NCH421K with NB3F18-Au^5nm^ bound to the plasma membrane, in the multivesicular body, and in the lysosome (arrows). (**C**) U87MG with NB3F18-Au^5nm^ bound to the plasma membrane and in a macropinosome/early endosome (arrows). (**D**) U251 cell line with NB3F18-Au^5nm^ bound to the plasma membrane as well as inside endocytic vesicles (arrows). (**E**) Astrocyte cell line with NB3F18-Au^5nm^ bound to the plasma membrane (arrows). (**F**) Control cell line U251MG without gold particles. (**G**) U87MG and (**H**) U251MG displaying the early stages of NB3F18-Au^5nm^ internalization. Higher magnification of areas in red frames are shown in black frames. Scale bars correspond to 600 nm.

**Figure 10 antibodies-14-00008-f010:**
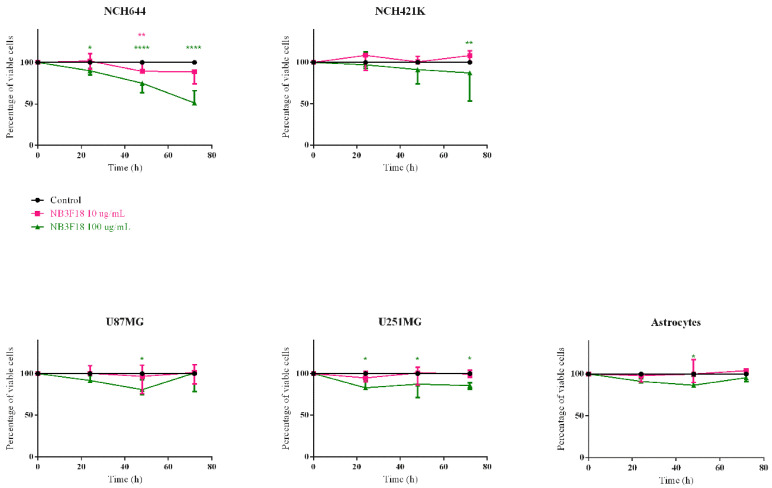
Effect of NB3F18 on the viability of glioblastoma cells and astrocytes. The WST-1 assay was used. * *p* ≤ 0.05; ** *p* < 0.01; **** *p* < 0.0001. The greatest difference was observed for NCH644 after 48 h and 72 h incubation with 100 µg/mL NB3F18 with **** *p* < 0.001 in both cases and 72 h incubation with 10 µg/mL NB3F18 with ** *p* = 0.0038.

**Figure 11 antibodies-14-00008-f011:**
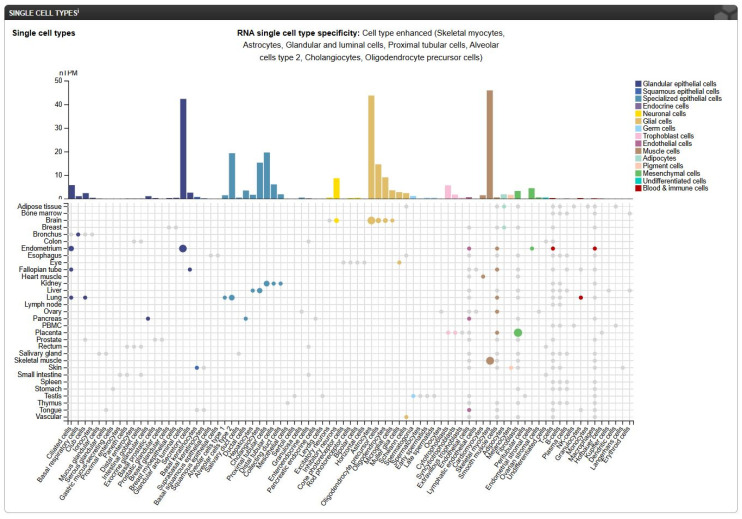
RNA single-cell type specificity. nTPM—normalized transcripts per million. Image credit: The Human Protein Atlas, https://www.proteinatlas.org/ENSG00000150893-FREM2/single+cell, access date 13 January 2025.

**Figure 12 antibodies-14-00008-f012:**
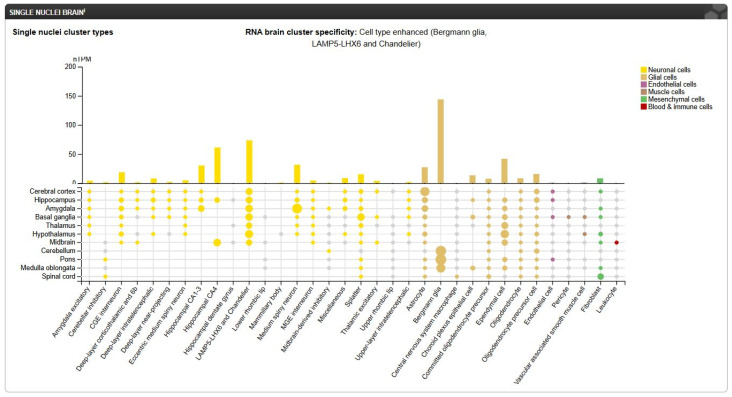
RNA brain cluster specificity. nTPM—normalized transcripts per million. Image credit: The Human Protein Atlas, https://www.proteinatlas.org/ENSG00000150893-FREM2/single+cell, accessed on 13 January 2025.

**Figure 13 antibodies-14-00008-f013:**
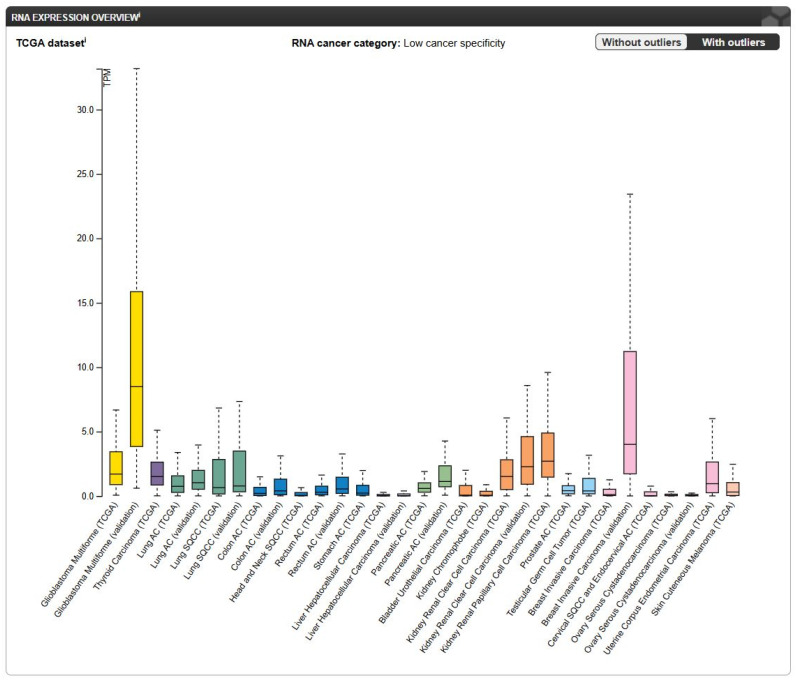
Overview of *FREM2* RNA expression levels in cancers analyzed in TCGA and experimental validation datasets. TPM—transcripts per million. Image credit: The Human Protein Atlas, https://www.proteinatlas.org/ENSG00000150893-FREM2/cancer, accesses on January 13, 2025.

**Figure 14 antibodies-14-00008-f014:**
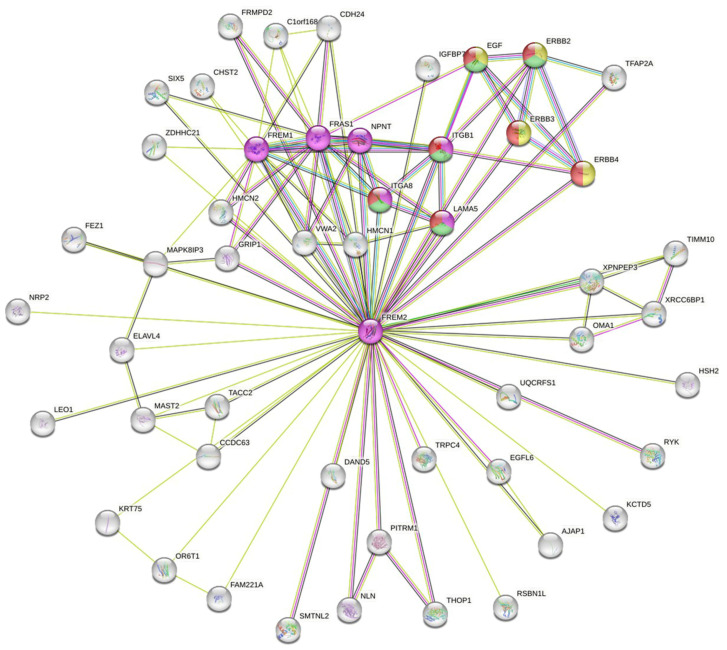
STRING network of FREM2. No more than 50 interactors in the first shell were included in the analysis. The following KEGG pathways are colored: ECM–receptor interaction (violet), ErbB signaling pathway (yellow), focal adhesion (green), and PI3K-Akt signaling pathway (red).

**Table 1 antibodies-14-00008-t001:** Computed parameters about NB3F18.

NB3F18	
Number of amino acids	122
Molecular weight	13,598.27 Da
Theoretical pI	9.01
Extinction coefficient (Abs 0.1% = 1 g/L)	1.476, assuming all Cys residues form cystines 1.466, assuming all Cys residues are reduced

**Table 2 antibodies-14-00008-t002:** Flow cytometry results. Percentage of gated cells treated with NB3F18 and anti-FREM2 antibody and untreated cells. SD—standard deviation; SEM—standard error of mean; CV—coefficient of variation.

	% Gated Cells	SD	SEM	CV%
	NCH644			
Untreated	1.998	0.885	0.511	25.581
NB3F18 treated	65.358	4.179	2.412	3.691
Untreated	3.140	1.909	0.854	27.198
Anti-FREM2 treated	58.798	20.805	9.304	15.824
	NCH421K			
Untreated	2.035	0.823	0.336	16.511
NB3F18 treated	36.474	11.044	4.939	13.541
Untreated	1.932	0.172	0.086	4.458
Anti-FREM2 treated	65.825	29.575	20.913	37.374
	U251MG			
Untreated	2.835	0.639	0.319	11.277
NB3F18 treated	19.037	5.441	2.720	14.291
Untreated	2.449	1.542	0.771	31.499
Anti-FREM2 treated	13.106	6.315	3.157	24.092
	U87MG			
Untreated	2.515	1.034	0.517	20.556
NB3F18 treated	25.251	6.059	3.029	11.998
Untreated	2.326	1.293	0.578	24.860
Anti-FREM2 treated	20.353	10.291	4.602	22.613
	Astrocytes			
Untreated	3.241	1.193	0.689	21.260
NB3F18 treated	25.348	10.463	6.041	23.833
Untreated	1.975	0.544	0.385	19.493
Anti-FREM2 treated	39.475	13.031	9.215	23.343

**Table 3 antibodies-14-00008-t003:** Statistical analysis of the cell proliferation assay. Table shows difference in cell proliferation between treated cells and non-treated controls at two different nanobody concentrations. * *p* ≤ 0.05; ** *p* < 0.01; **** *p* < 0.0001, ns—nonsignificant.

	Time of Incubation					
	24 h		48 h		72 h	
	**c (NB3F18) µg/mL**					
	10	100	10	100	10	100
NCH644	ns	ns	ns	**** *p* < 0.001	** *p* = 0.0038	**** *p* < 0.001
NCH421K	ns	ns	ns	ns	ns	** *p* = 0.0031
U87MG	ns	ns	ns	* *p* = 0.0221	ns	ns
U251MG	ns	* *p* = 0.0137	ns	* *p* = 0.0346	ns	* *p* = 0.0157
Astrocytes	ns	ns	ns	* *p* = 0.0256	ns	ns

## Data Availability

The original contributions presented in this study are included in the article/Appendix A. Further inquiries can be directed to the corresponding authors.

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
