# Peer review of "In Vitro Functional Validation of an Anti-FREM2 Nanobody for Glioblastoma Cell Targeting"

_2073-4468, 2025, doi:10.3390/antib14010008_

Round 1
Reviewer 1 Report
Comments and Suggestions for Authors
The authors have demonstrated the potential of a nanobody, NB3F18, to target FREM2 on the surface of glioblastoma cells. This work, titled "In vitro functional validation of an anti-FREM2 nanobody for glioblastoma cell targeting", is commendable. The nanobody was successfully identified through screening, and its various properties were thoroughly evaluated. This study is promising, and I hope the authors will continue to explore this avenue in the future. However, I have a few questions and suggestions that should be addressed in the manuscript:
1. Could you consider introducing the AlphaFold 3 model into the manuscript, if feasible? Currently, it is unclear whether molecular dynamics (MD) simulations outperform AlphaFold 3 for such analyses. This comparison could provide additional insights and strengthen the study.
2. It would be valuable to include data on the distribution of FREM2 in different types of glioblastoma stem cells. Incorporating this information into Figure 5 would provide a more comprehensive understanding of the nanobody's targeting potential across various cell subtypes.
3. In Figure 6, I observed that NB3F18 can enter U251MG, U87MG, and astrocyte cells but not NCH644 and NCH421K cells. Could this suggest that at least two different receptors are involved in mediating NB3F18’s interaction with the cell surface? One of these receptors might specifically facilitate cell entry for NB3F18. It would be interesting to explore this hypothesis in future experiments.
Overall, this is a well-executed and promising study, and I encourage the authors to continue this important line of research.
Author Response
Comment 1. Could you consider introducing the AlphaFold 3 model into the manuscript, if feasible? Currently, it is unclearwhether molecular dynamics (MD) simulations outperformAlphaFold 3 for such analyses. This comparison could provide additional insights and strengthen the study. Thank you for the suggestion.Please see newly introduced section 2.8 Modeling the 3D interaction between nanobody NB3F18 and human FREM2 with AlphaFold3.0 (Materials and methods), which as well as 3.4 AlphaFold3.0 shows low prediction accuracy for FREM2 protein and in simulating its interactions with NB3F18 (Figure 6 in Results). We have also updated the discussion.
Our findings are in line with the low confidence for predicted structure of FREM2 which is also reported in The Human Protein Atlas (https://www.proteinatlas.org/ENSG00000150893-FREM2/structure+interaction#interaction).
Comment 2: It would be valuable to include data on the distribution of FREM2 in different types of glioblastoma stem cells. Incorporating this information into Figure 5 would provide a more comprehensive understanding of the nanobody's targeting potential across various cell subtypes
We thank the reviewer for this suggestion. However, we think this information is better suited in the Discussion so please see the revised discussion and newly added Figures 11, 12 and 13.
When it comes to the expression of FREM2 in different types of glioblastoma cells (eg. stem cells versus differentiated cells) we are afraid we cannot provide such results as information is unavailable. The Human Protein Atlas contains only expression data about differentiated cells (image below, https://www.proteinatlas.org/ENSG00000150893-FREM2/cell+line). If the reviewer or editor considers this information is relevant we can add it to the manuscript. Figures available in the revised text)
Comment 3: In Figure 6, I observed that NB3F18 can enter U251MG, U87MG, and astrocyte cells but not NCH644 and NCH421K cells. Could this suggest that at least two different receptors are involved in mediating NB3F18’s interaction with the cell surface? One of these receptors might specifically facilitate cell entry for NB3F18. It would be interesting to explore this hypothesis in future experiments.
It is written in section 3.7 that NB3F18-546Alexa localizes to the surface of glioblastoma stem cells NCH644: “besides binding intracellularly, NB3F18 (red) bound to the surface of glioblastoma stem cells NCH644 and NCH421K (white arrows), while its localization in differentiated glioblastoma cells U251MG and U87MG, and astrocytes was cytoplasmic”. We would like to point out that besides its presence on the cell surface, NB3F18 was also detected intracellularly in NCH644 and NCH421K. However, this was not observed in the case of differentiated U251MG and U87MG glioblastoma cells nor in astrocytes.
Since we have not performed experiments to determine the binding and entry mechanism of NB3F18 to glioblastoma cells and astrocytes we cannot comment on the involvement of potential binding and entry receptors. However, we thank the reviewer for this suggestion and will consider performing such experiments in our future research
Reviewer 2 Report
Comments and Suggestions for Authors
The authors here discovered a potent nanobody against FREM2 enriched in glioblastoma cells. Overall, this work is well done and can be considered for publication.
1. In the Introduction part, the description of FREM2 function is relatively less, although the first paragraph in the Discussion part is provided. Maybe these contents can be reorganized. The summary of the important conclusion and the possible limitations can be more focused in the Discussion part.
2. The procedure of nanobody preparation can be provided as one scheme figure. It is a very important part in this paper.
3. There is some difference in the subcellular localization of NB3F18 in different types of cells. Some possible reasons can be also provided.
4. Is cell selectivity of NB3F18 also related to the expression level of FREM2 in different cells?
5. Is the function of FREM2 enough important to show the cytotoxicity when FREM2 is blocked by the nanobody? The cell viability results for these cells types are showing different. FREM2 as antitumor target or the receptor-mediated endocytosis for the delivery other functional molecules?
6. The plasmid sequence encoding NB3F18 for bacterial expression can be added as a supplementary table. Is there a SDS-PAGE gel validation of the purified NB3F18?
Author Response
Comment 1: In the Introduction part, the description of FREM2 function is relatively less, although the first paragraph in the Discussionpart is provided. Maybe these contents can be reorganized. The summary of the important conclusion and the possible limitations can be more focused in the Discussion.
Answer: We agree. As suggested, we reorganized parts of the Introduction and Discussion sections. We have decided to keep the Conclusion section as it was.
Comment 2: The procedure of nanobody preparation can be provided as one scheme figure. It is a very important part in this paper.
Answer: This was done, please consult new Figure 1.
Comment 3: There is some difference in the subcellular localization of NB3F18 in different types of cells. Some possible reasons can be also provided.
Answer: The exact reasons for the differences in subcellular localization of NB3F18 remain unclear. We shoudl not forget that selections of Nanobodies was perforemd on a small fraction of FREM2 protein. It remains possible that on cells this part interacts with other proteins or other extracellular domains and that the in vivo (affinity matured) heavy chain-only antibody antibody recognizes FREM2 plus something else. This might alos explain the low affinity of the Nanobody interaction to the FREM2 polypeptide. Other potential reasons for differences in subcellular locatlization of the NB3F18 are different isoforms, post-translational modifications, interactions with other proteins. Additionally, specific signaling pathways unique to the cell type could play a role in determining the subcellular compartmentalization of FREM2. As implicated in the immunofluorescence and TEM experiments, FREM2 may appear with different localization depending on the cell type. This can be a result of variations in the endocytic machinery, different growth dynamics of the cells and also expression of markers other cell-type-specific molecules. The different subcellular localization of NB3F18 in various cell types may also be a result of cell-specific internalization mechanism. In conclusion, we believe that at this stage, too detailed speculation about the different subcellular localizations is premature.
Comment 4: Is cell selectivity of NB3F18 also related to the expression level of FREM2 in different cells?
Answer:
The higher expression levels of FREM2 in NCH cells compared to U251MG, U87MG and astrocytes were published in our previous publication (https://www.mdpi.com/1422-0067/19/5/1369). This was also confirmed with the flow cytometry experiment in this manuscript where NB3F18 showed more important binding to NCH644 and NCH421K cells compared to U251MG and U87MG cells as well as astrocytes. Furthermore, because of a higher FREM2 expression level, the cytotoxic effect of NB3F18 on GSCs NCH644 and NCH421K may be related to their higher metabolic activity consistent with their role in maintaining stemness. Comment 5: Is the function of FREM2 enough important to show the cytotoxicity when FREM2 is blocked by the nanobody? The cell viability results for these cells types are showing different. FREM2 as antitumor target or the receptor-mediated endocytosis for the delivery other functional molecules? Answer:Our STRING analysis links FREM2 to ECM-receptor interactions and PI3K/Akt signaling which implies its importance in cell survival and motility. We have shown that NB3F18 has a more pronounced effect on GSCs compared to differentiated cells as well as astrocytes. Since GSCs are known to be resistant to chemotherapy leading to tumor relapse, the nanobody could be used as a therapeutic agent. Additionally, NB3F18 can serve as a delivery vehicle for other cytotoxic molecules, although further evaluation is required to explore this possibility.
Comment 6: The plasmid sequence encoding NB3F18 for bacterial expression can be added as a supplementary table. Is there a SDS-PAGE gel validation of the purified NB3F18?
Answer:
The plasmid sequence encoding NB3F18 for bacterial expression is provided schematically in reference 51 and whole sequencxe can be obtained from us upon request.
Yes, Supplementary file 2 is the SEC chromatogram of NB3F18 purification and elution, while Supplementary file 3 is SDS-PAGE of NB3F18 after large-scale expression and purification
Reviewer 3 Report
Comments and Suggestions for Authors
Authors have mentioned the detailed protocol for the development and selection of the anti-FREM2 nanobody for targeting gioblastoma cells.
Authors can provide more details on the below-mentioned comments so it can increase the reproducibility and reader understanding
· In the Introduction, the author can discuss the FDA approved nanobodies such as Caplacizumab and any others to elaborate on the development of nanobody therapeutic potential.
· If the biomarkers (CD133, EGFR) in the glioblastomas have therapeutic potential, please explain.
· Authors can provide more details about the immunization protocol (lane 123-). Source of the cells, adjuvant used? Frequency of the immunization and other details
· The Concentration of FREM2 (100 μg/mL) for immobilization is optimized or mentioned reference?
· Lane 137: what is the final concentration of IPTG
· Lane 158: Source of the plasmid pHEN4 and pHEN6.
· Please mention the labeling efficiency of NB3F18-546 Alexa after labeling and how the functional integrity of NB3F18-546 Alexa was confirmed post-labeling.
· Please mention the initial and final concentration of NB3F18-546Alexa after desalting.
· In Flowcytometry please mention the controls used like unstained cells, secondary antibody-only controls, or isotype controls.
· Please mention if the Cell viability was measured before and after flow cytometry to confirm the binding was not compromised with cell viability
· Please provide more details on why the specific FREM2 fragment (residues 2325–2451) was chosen.
· Please mention additional details on the criteria used to select NB3F18 over NB2F111 for further analysis, beyond the SPR results.
· Please explain the mechanisms driving NB3F18-Au5nm internalization (lane 514).
· Please mention the viability of astrocytes to strengthen the non-tumor targeting criteria.
Author Response
Comment 1: In the Introduction, the author can discuss the FDA approved nanobodies such as Caplacizumab and any others to elaborate on the development of nanobody therapeutic potential.
Answer: We agree. Please see revised Introduction
Comment 2: If the biomarkers (CD133, EGFR) in the glioblastomas have therapeutic potential, please explain.
Answer: We agree, This is now added in the Introduction.
Comment 3: Authors can provide more details about the immunization protocol (lane123-). Source of the cells, adjuvant used? Frequency of the immunization and other details
Answer: We agree, the M&M was extended to include additional information. Also details of the panning are provided.
Comment 4: The Concentration of FREM2 (100 μg/mL) for immobilization is optimized or mentioned reference.
Answer: We agree:
Please see revised In section "2.3 Library enrichment and screening" (in Materials and methods) we provided details and added references.
Comment 5: Line 137: what is the final concentration of IPTG
Answer: The final concentration is 1 mM, this is now added.
Comment 6: Line 158: Source of the plasmid pHEN4 and pHEN6
Answer: Plasmids pHEN4 and pHEN6 were generated in house by the group of Serge Muyldermans at Vrije Universiteit Brussel. Please see 2.5 Subcloning, large-scale expression and purification in Materials and methods A scheme of the pHEN4 vector was published in citation 51. The entire sequence can be provided by us upon simple request. We can even provide the cloning vector for free, if needed.
Comment 7: Please mention the labeling efficiency of NB3F18-546 Alexa after labeling and how the functional integrity of NB3F18-546 Alexa was confirmed post-labeling.
Answer: We agree. Please see revised "3.5 NB3F18 labelling with Alexa Fluor 546 dye" in Results section
Comment 8: Please mention the initial and final concentration of NB3F18-546Alexa after desalting.
Answer: We agree and it was done. Please see revised "3.5 NB3F18 labelling with Alexa Fluor 546 dye" in Results section.
Comment 9: In Flow cytometry please mention the controls used like unstained cells, secondary antibody-only controls, or isotype controls.
Answer: We agree. Please see revised "2.10 Flow cytometry in Materials and methods" and "3.6 Cellular specificity of NB3F18 towards glioblastoma stem cells NCH644" in Results section.
Comment 10: Please mention if the Cell viability was measured before and after flow cytometry to confirm the binding was not compromised with cell viability.
Answer: We did not examine cell viability after flow cytometry, because we had previously performed a cell proliferation assay. Furthermore, the cells appeared healthy under the microscope, and the dot plot graphs showed a single coherent population of cells.
Comment 11: Please provide more details on why the specific FREM2 fragment (residues 2325–2451) was chosen.
Answer: A good question. According to the UniProt protein database which uses the code Q5SZK8 for FREM2 the FREM2 peptide that we used in our studies belongs to the extracellular part of the FREM2 protein, starting at amino acid residue 2325 and ending at amino acid residue 2451 (section Subcellular location, subsection Topology, and section Sequences (2), Isoform 1). Due to the membrane-associated localization of FREM2 and its overexpression in glioblastomas, the FREM2-specific nanobody can be used for selective targeting of glioblastoma cells with minimal damage to adjacent cells and without the need for entering the cell. Hence, it was important that the nanobody interacted with the extracellular part of FREM2.
Here we report the isolation, characterization and functional validation of a nanobody, NB3F18, directed against the corresponding protein of FREM2. Due to the membrane-associated localization of FREM2 and its overexpression in glioblastomas, the FREM2-specific nanobody can be used for selective targeting of glioblastoma cells with minimal damage to adjacent cells and without the need for entering the cell. Hence, it was important that the nanobody interacted with the extracellular part of FREM2.
This information is given in "3.1 Identification of an anti-FREM2 nanobody and its preparation for in vitro use" in Results and in the Discussion.
Comment 12: Please mention additional details on the criteria used to select NB3F18 over NB2F111 for further analysis, beyond the SPR results.
Answer: Both NB2F111 and NB3F18 were selected as potential binders because they showed at least 10× higher signal in wells coated with FREM2 peptide compared to wells without peptide. In addition, both nanobodies had the complete nanobody sequence with the characteristic amino acid sequences at the amino-terminus (i.e., QVQL) and carboxy-terminus (i.e., TVSS), as well as distinct CDR1, CDR2 and CDR3. However, preliminary SPR showed that only NB3F18 binds to the FREM2 peptide and at this point we excluded NB2F111 from further analysis. Additional tests were not performed.
Comment 13: Please explain the mechanisms driving NB3F18-Au 5nm internalization (line 514).
Answer: The proposed mechanism of internalization is based on ultrastructural criteria (Doherty GJ, McMahon HT. Mechanisms of endocytosis. Annu Rev Biochem 2009:78:857-902. doi: 10.1146/annurev.biochem.78.081307.110540, Juliano RL, Ming X, Nakagawa O. Bioconjug Chem 2012 Feb 15;23(2):147-57. doi: 10.1021/bc200377d). To reliably determine the mechanism of internalization immunolabeling eg. of clathrin, small G-proteins, dynamin and other implicated proteins would be required.
Comment 14: Please mention the viability of astrocytes to strengthen the non-tumor targeting criteria.
Answer: We agree: Please see revised Discussion and Conclusion.
Round 2
Reviewer 2 Report
Comments and Suggestions for Authors
The authors have well addressed all my concerns. This manuscript can be accepted for publication in its present form.